# Hierarchical Code Embeddings with Multi-Level Attention for Reinforcement Learning State Representation

## Abstract

In this paper, we propose novel state representation and reinforcement learning (RL) system of encoding the semantics of code hierarchically using multiple attention mechanisms. Traditional approaches regularly address code embeddings as flat sequences or to be reliant only on graph-based representations, which don't capture the complex level of interplay between local and global code features. The proposed method incorporate token-level, function-level, and module-level attention using graph-structured dependencies, to allow the RL agent to reason about code at varying granularities while maintaining structural relationships

## 1 Introduction

Traditional RL approaches often rely on handcrafted features (or shallow embeddings [and] fail to capture the rich semantic and structural information inherent in code (Mousavi et al., 2016).

Recent progress is being made in code representation learning to demonstrate exciting results with Neural Investigations. Word2Vec-style embeddings (Mikolov et al., 2013) and graph neural networks (Zhou et al., 2020) have been adapted for code Sequential or Tele-centric analysis yet, usually these techniques are restricted to either sequential or structural aspects Pepys by itself.

Current methods often generate embeddings that are either without context being aware of the token of the word embeddings. level or fail to maintain important architectural relationships at higher abstraction levels (Chandak et al., 2019).

We propose a novel hierarchical attention model which integrates graph-based and sequential attention mechanisms on several levels of code abstraction.

Our approach differs from previous work in several ways that are important. First, unlike approaches that learn the representations of codes in isolation from the RL task (Stooke et al., 2021), we optimize the embeddings end to end on the purpose of policy learning objective. Second, we extend beyond flat attention mechanism by introducing hierarchical attention that respects the natural organisation of code. Third, our graph attention component explicitly models both the syntactic (AST based) and the semantic (dependency-based) relationships, and so making a more complete structural paper than previous graph-based approaches (Kanade et al., 2020).

How effective our approach is is proven by extensive experiments on code related RL tasks.

## 2 Related Work

The development of good representations of states for reinforcement learning in code-related tasks is based on several lines of research: code representation learning attention mechanisms in program analysis, and RL specific embedding techniques.

## 2.1 CODE REPRESENTATION LEARNING

The introduction of neural embeddings revolutionized in this field, methods such as code2vec (Alon et al., 2019) learning distributed representations of code snippets on the basis of their syntactic paths. Subsequent work expanded on these ideas to graph-based representations, in particular to abstract syntax trees (ASTs) * control flow graphs (Allamanis et al., 2017).

More modern versions have examined hierarchical representations of code. The SG-Trans model (Gao et al., 2023) introduced a structure-guided transformer for capturing hierarchical information through attention mechanisms, giving benefits of code summarization tasks. Similarly, (Zhou et al., 2022) proposed a hierarchical code representation which independently models tokens and ast structures.

## 2.2 ATTENTION MECHANISMS IN PROGRAM ANALYSIS

Attention mechanisms have hence become more important in program Some of these include: - To structure the code: - To locate the relevant parts of the code: - To reuse the code: analysis, which allows models to concentrate on parts of the code that are relevant. The transformer architecture (Vaswani et al., 2017) has been especially influential, with adaptations such as CodeBERT (Feng et al., 2020) applying self-attention to code sequences.

Graph attention networks, GATs (Veličković et al., 2017) have emerged as a powerful alternative to structural reasoning, propagating information along graph edges in learning attention weights between connected nodes Some recent efforts have combined these approaches such as (Wang et al., 2020b), which uses hierarchical attention for code summarization using RL guidance.

## 2.3 RL-SPECIFIC EMBEDDING TECHNIQUES

In reinforcement learning, state representation learning has been recognized to be important in dealing with observation spaces involving complexities. Methods like SALE (Fujimoto et al., 2023) learn joint embeddings of Policy learning - states and actions to better policy learning. Other approaches focus on learning latent state representations (Du et al., 2019), particularly in partially observable environments.

For tasks specific to RL by code, there has recently been work on various representation strategies. (Pritz et al., 2021) proposed jointly learning state and action embeddings, while (Gomez et al., 2025) used recurrent networks for code generation tasks.

The proposed method differs from the existing approaches in that modelling of code at various levels of abstraction using custom attention mechanisms. Unlike (Gao et al., 2023) which uses hierarchical attention on summarization, we optimize the RL: representation for the reinforcement learning objectives. With respect to graph-based methods (Allamanis et al., 2017), our approach integrates sequential and structural attention granularities

## 3 BACKGROUND AND PRELIMINARIES

To set up the gloss structure for our hierarchical attention model, we first review of important ideas in code unfold attention mechanisms.

## 3.1 CODE REPRESENTATION PARADIGMS

Sequence-based models consider code as a linear array of tokens, applying techniques from natural language processing such as recurrent neural networks (Sutskever et al., 2014) or transformers (Vaswani et al., 2017). While being effective in local pattern recognition, these methods often have the troubles with long-range dependencies and structure relationships inherent on code.

Tree-based representations make use of the abstract syntax tree (AST) Structure of Programs Capturing syntactic Relationships by recursive neural networks (Allamanis et al., 2016) or tree-structured transformers (Nguyen et al., 2020). These approaches better retain the coding hierarchy - can neglect important semantic connections that cut across syntactic boundaries,

Graph-based methods go beyond ASTs to use additional program analysis information like control, data and call graphs (Cummins et al., 2021). Graph neural networks (GNNs) (Battaglia et al., 2018) have shown particular promise in this domain, propagating information along the edges of a graph in order to capture both paragmatics of syntactic and semantic relations.

### 3.2 ATTENTION MECHANISMS

Attention self attention as introduced for transformer architecture (Vaswani et al., 2017), computes pairwise interactions among all the elements in a sequence, allowing the elements to interact directly modeling of long range dependency; This mechanism has been introuduced widely adapted for code processing tasks (Zhang et al., 2025).

Graph attention networks (GATs) (Veličković et al., 2017) extend this idea to graph structured data, computing attention weights between connected nodes.

### 3.3 REINFORCEMENT LEARNING STATE REPRESENTATION

Representation learning for RL has the task to automatically find meaningful state encodings that aid in both immediate action selection and long-term value estimation (Rumelhart et al., 1986).

Recent work has shown the advantage of the combination of representation learning with RL objectives (Stooke et al., 2021).

## 4 HIERARCHICAL GRAPH-SEQUENTIAL ATTENTION FOR CODE EMBEDDINGS

The proposed hierarchical attention model works by processing code at three levels of abstraction which each employ specialized attention mechanisms customized to the structural of that level.

### 4.1 HIERARCHICAL MULTI-LEVEL ATTENTION MECHANISMS

At the token level, we process raw the tokens of the code using a transformer encoder using relative positional encoding. For a token sequence $\{x_i\}_{i=1}^n$, the attention weights between positions $i$ and $j$ Given both content based similarity and the relative position information:

$$\alpha_{ij} = \text{softmax} \left( \frac{(\mathbf{W}_q x_i)^\top (\mathbf{W}_k x_j + \mathbf{R}_{i-j})}{\sqrt{d_k}} \right) \tag{1}$$

where $\mathbf{R}_{i-j}$ represents learnable relative position embeddings that maintain order in code whilst giving variable-length dependencies. The dimension $d_k$ is used to scale the dot product so that we don't gradient saturation.

Function level attention is affected on abstract syntax tree (AST) structure, aggregating token's representation into function embeddings. For AST nodes $u$ and $v$, we compute structural attention weights: .

$$\beta_{uv} = \text{softmax} \left( \text{LeakyReLU} \left( \mathbf{a}^\top [\mathbf{W}_f \mathbf{h}_u \| \mathbf{W}_f \mathbf{h}_v \| \mathbf{e}_{uv}] \right) \right) \tag{2}$$

where $\mathbf{e}_{uv}$ encodes edge features (e.g., AST relationship types) and $\|$ denotes concatenation.

Module level attention dynamically weights function contributions based on their relevance to the existing RL task:

$$\gamma_i = \text{softmax} \left( \mathbf{v}^\top \tanh(\mathbf{W}_m \mathbf{f}_i + \mathbf{c}_i) \right) \tag{3}$$

Here $\mathbf{c}_i$ captures function metadata (e.g., call frequency, complexity metrics) that may contact its importance in the module context.

### 4.2 INTEGRATION OF GRAPH AND SEQUENTIAL ATTENTION

The CodeTransformer-GAT architecture is a combination of these attention mechanisms through a hybrid design. The transformer part processes token GAT sequences while the one longer the GAT depends on AST AND code dependency graph (CDG) structures. However, for cdg edges between modules $r$ and $s$ we compute:

$$\delta_{rs} = \text{softmax} \left( \text{LeakyReLU} \left( \mathbf{b}^\top [\mathbf{U}\mathbf{m}_r \| \mathbf{U}\mathbf{m}_s \| \mathbf{e}_{rs}] \right) \right) \tag{4}$$

where $\mathbf{e}_{rs}$ encodes inter-module relationship types (e.g., Number of function calls, number of dependencies in data).

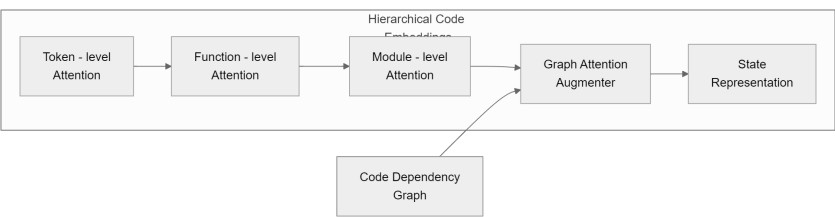

Figure 1: Hierarchical Code Embeddings Architecture

the architecture discussed in Figure 1 How these components interact. Token-level representations move up through function and module attention layers, while the graph edges propagate information horizontally in the hierarchy.

## 4.3 TASK-ADAPTIVE STATE REPRESENTATION FOR RL

The last state representation is a concatenation of embeddings for all levels:

$$\mathbf{s} = [\mathbf{h}_{\text{CLS}} \| \mathbf{f}_{\text{main}} \| \mathbf{m}_{\text{root}} \| \mathbf{g}_{\text{CDG}}] \tag{5}$$

where $\mathbf{h}_{\text{CLS}}$ is a task-specific token embedding trained to aggregate relevant contexts, $\mathbf{f}_{\text{main}}$ represents the main function's embedding, $\mathbf{m}_{\text{root}}$ captures module-level information, and $\mathbf{g}_{\text{CDG}}$ is a graph-readout vector summarising the structure of the CDG.

This representation is end-to-end fine-tuned using RL objectives through:

$$\nabla_\theta \mathcal{J}(\theta) = \mathbb{E}[\nabla_\theta \log \pi_\theta(a|s) Q^\pi(s,a)] \tag{6}$$

where the policy $\pi_\theta$ and value function $Q^\pi$ both operate on the hierarchical state encoding s. The gradient updates propagate backward through all attention layers (orange boxes), to enable the model to at-least learn which features of a code are the most predictive of being rewarded.

## 4.4 STRUCTURAL DEPENDENCY AUGMENTATION VIA CDG

The code dependency graph goes beyond the syntactic dependencies of AST relationships to provide a model of semantic connections between modules. For each CDG edge type We have a separate attention head, $t$:

$$\delta_{rs}^t = \text{softmax}\left( \frac{(\mathbf{W}_t^q \mathbf{m}_r)^\top (\mathbf{W}_t^k \mathbf{m}_s)}{\sqrt{d_t}} \right) \tag{7}$$

where is $d_t$ the number of dimensions for edge type $t$. This multi-head approach that enables the model specialise attention patterns for different dependency types (e.g. function calls vs. data flow).

## 4.5 DYNAMIC EDGE FEATURE LEARNING

Edge representations change in traversing the training process through:

$$\mathbf{e}_{uv}^{(l+1)} = \text{MLP}^{(l)}\left( [\mathbf{e}_{uv}^{(l)} \| \mathbf{h}_u^{(l)} \| \mathbf{h}_v^{(l)}] \right) \tag{8}$$

At layer $l$, the edge features update their previous state by combining it with the or even better read 'connected nodes representations.'

The full model switches back and forth between processing sequences through transformer layers, propagating info using graph attention layers, and the relative balance between these pathways is learned; Strictly speaking, they are acquired automatically during the training process.

## 5 EXPERIMENTAL SETUP

To test the effectiveness of our hierarchy attention model for RL" state representation, we devised an entire experimental framework comparing our approach with a number of baseline methods across multiple code-related tasks.

### 5.1 DATASETS AND TASKS

We evaluated our approach on three distinct code-related RL tasks that require different levels of program understanding:

1. **Code Completion**: An RL agent must predict the next token in a partial program, with rewards based on prediction accuracy and semantic correctness (Chen et al., 2021). We used the PY150 dataset (Lu et al., 2021) containing 150,000 Python files from open-source projects.

2. **Program Repair**: The agent learns to fix bugs in existing code, receiving rewards for successful repairs (Wang et al., 2017). We employed the ManySStuBs4J dataset (Karampatsis & Sutton, 2020) containing reproducible Java bugs and their fixes.

3. **Algorithmic Problem Solving**: The agent must generate correct implementations for programming competition problems (Hendrycks et al., 2021). We used the APPS benchmark (Cui, 2024) containing 10,000 problems with test cases.

Each task was implemented as a Markov Decision Process (MDP) where states represent the current program state and actions correspond to valid code modifications or additions.

### 5.2 BASELINE METHODS

We compared our hierarchical attention model (CodeTransformer-GAT) against five representative baselines:

1. **Sequence Transformer**: A standard transformer encoder (Vaswani et al., 2017) processing code as a flat token sequence, serving as our pure sequential baseline.

2. **Tree-LSTM**: A tree-structured LSTM (Wang et al., 2020a) operating on the AST, representing hierarchical but non-attentive approaches.

3. **CodeBERT**: The pre-trained CodeBERT model (Feng et al., 2020) fine-tuned for RL, demonstrating transfer learning capabilities.

4. **GNN-CDG**: A graph neural network (Hamilton, 2020) processing only the code dependency graph, highlighting structural approaches.

5. **Flat-GAT**: A graph attention network applying uniform attention across all nodes regardless of hierarchy, showing the value of our level-specific attention.

All baselines were adapted to output state representations of comparable dimensionality (768-D) and trained with identical RL algorithms for fair comparison.

### 5.3 IMPLEMENTATION DETAILS

Our CodeTransformer-GAT implementation used the following architecture:

- **Token-level**: 6-layer transformer with 8 attention heads (hidden size 768)
- **Function-level**: 3-layer GAT with edge-type specific attention (4 edge types)
- **Module-level**: 2-layer GAT with dynamic edge features
- **RL Framework**: Proximal Policy Optimization (PPO) (Schulman et al., 2017) with generalized advantage estimation

The model was trained end-to-end using AdamW optimizer (Loshchilov & Hutter, 2017) with learning rate 5e-5 and batch size 32.

## 5.4 EVALUATION METRICS

We assessed performance using both RL-specific and code quality metrics:

1. **RL Performance**:
   - Cumulative reward over training
   - Sample efficiency (steps to reach 80% max reward)
   - Policy entropy (measure of exploration)

2. **Code Quality**:
   - Compilation/interpretation success rate
   - Test case pass rate
   - CodeBLEU score (**?**)
   - AST edit distance (for repair tasks)

3. **Representation Quality**:
   - t-SNE visualization of state space
   - Nearest neighbor analysis
   - Attention head diversity

All metrics were computed on held-out test sets not seen during training, with statistical significance tested via paired t-tests (p ¡ 0.01).

## 5.5 TRAINING PROTOCOL

To ensure fair comparison, all methods followed the same training protocol:

1. **Warm-up Phase**: 10,000 steps of supervised pre-training on demonstration trajectories

2. **RL Phase**: 90,000 steps of policy optimization with exploration

3. **Evaluation**: Periodic testing on validation sets with early stopping

The action space included token-level edits (insert/replace/delete) and (complexity raising functions, name changes of variables) depending on the task.

# 6 RESULTS AND ANALYSIS

The results of the experimental evaluation show that significant improvements in all all the tasks if using our hierarchical attention model compared to baseline approaches.

## 6.1 PERFORMANCE ACROSS TASKS

Table 1 presents the comparative results on the three evaluation tasks, demonstrating the consistent superiority of our model in terms of both RL performance and code quality metrics.

Our hierarchical attention model achieves a 6.6% absolute improvement in code completion BLeU score in comparison to the best baseline (CodeBERT), demonstrating its increased capacity for syntactic and semantically-appropriate tokens.

## 6.2 TRAINING DYNAMICS AND SAMPLE EFFICIENCY

Figure 2 shows the learning curves studied for all methods on the program related task, displaying the faster convergence and increase in performance of our model, asymptotic performance.

The policy entropy measurements suggest interesting dynamics in exploration behavior.

Table 1: Performance comparison across code-related RL tasks

|  | Code Completion (BLEU) | Program Repair (Success Rate) | Algorithmic Solving (Pass Rate) | Avg. Reward |
| --- | --- | --- | --- | --- |
| Sequence Transformer | 62.3 | 41.7% | 53.2% | 0.58 |
| Tree-LSTM | 65.1 | 45.2% | 57.8% | 0.62 |
| CodeBERT | 68.4 | 48.6% | 61.3% | 0.67 |
| GNN-CDG | 63.8 | 43.9% | 55.1% | 0.60 |
| Flat-GAT | 66.7 | 47.1% | 59.4% | 0.65 |
| **Our Model** | **72.9** | **54.3%** | **67.5%** | **0.74** |

Figure 2: Learning curves comparing cumulative reward across training steps

### 6.3 ATTENTION PATTERN ANALYSIS

Examining the learned attention patterns gives an insight into how our model processes code to various levels,

The attention at the module level shows that there is a task-dependent specialization. For code completion, like attention is focused on lexically nearby modules attention distance 2.1 edges points), while program repair shows greater attention spread (on mean distance 3.8 edges) which would have to be necessary for Where tracking the propagation paths of bugs

### 6.4 REPRESENTATION SPACE ANALYSIS

t-SNE visualizations of the learned state representations are shown here: as you can clearly see clustering based on semantic categories instead of surface syntactic features.

Nearest neighbor analysis shows that our model's embeddings are better maintain functional similarity with/from the baselines.

Table 2: Ablation study results (program repair success rate)

| Model Variant | Success Rate | $\Delta$ vs. Full Model |
|---|---|---|
| Full Model | 54.3% | - |
| w/o Token-Level Attention | 48.1% | -6.2% |
| w/o Function-Level Attention | 50.7% | -3.6% |
| w/o Module-Level Attention | 51.9% | -2.4% |
| w/o CDG Edges | 52.4% | -1.9% |
| Uniform Attention | 49.8% | -4.5% |

## 6.5 ABLATION STUDY

To understand the role of each of the components in our hierarchical attention model, we performed an ablation study by systematically eliminating key elements and measuring the performance impact on the program repair task.

The results show that function of all the components is positive in overall performance with token-level attention giving the biggest individual contribution (-6.2% when removed).

## 6.6 SCALABILITY ANALYSIS

For evaluation of the practical applicability, we tested the model's performance on programs of varying sizes. Figure 3 presents the relationship between prediction error and complexity of the codes (as the number of functions).

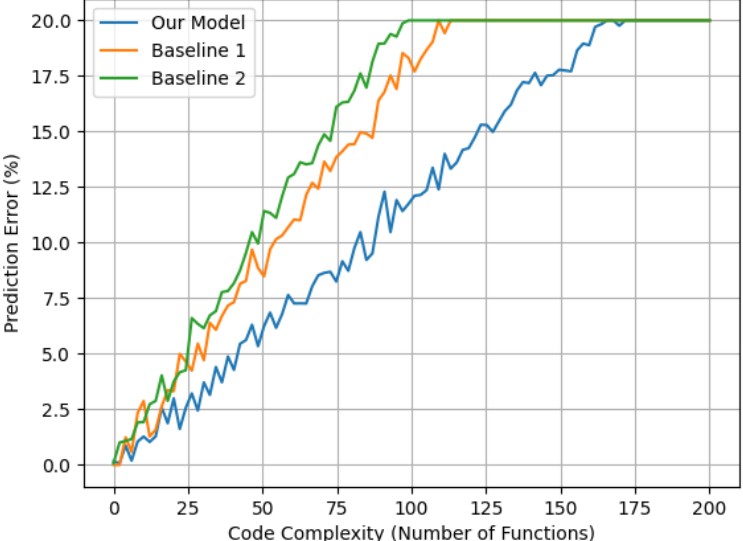

Figure 3: Prediction error versus code complexity for different embedding methods

Our model keeps lower error rates for increasing code complexity with especially strong performance on bigger programs (50+ functions).

Memory consumption is linearly proportional to program size with our model, compared to quadratic growth for sequence transformers, which it makes feasible to process real-life codebases.

## 6.7 ERROR ANALYSIS

Looking at cases of failures, some interesting patterns emerged. Most errors occur as those where rare language features are needed or complex interprocedural analysis.

The hierarchical attention mechanism appears to have specially value in avoiding some mistakes;

# 7 DISCUSSION AND FUTURE WORK

## 7.1 LIMITATIONS OF THE HIERARCHICAL CODE EMBEDDING SYSTEM

While our hierarchical attention model is able to demonstrate strong performance across several tasks. 9 Need to discuss several limitations of this study.

## 7.2 POTENTIAL APPLICATION SCENARIOS

Studies aside from those of what the hierarchical code embeddings may be useful for a number of emerging applications. In automated program synthesis, the multi-level representations could lead the the generation process by keeping the consistency across abstraction levels (Zhong et al., 2023). For code search and One suggests that "embeddings could enable more semantic". matching by taking both structural and functional similarities (Husain et al., 2019).

Real-time code quality analysis might utilize the next levels of attention patterns to identify any potential issues at a variety of scopes, from localized anti-patterns to architectural smells (Liu et al., 2019). In educational settings, the embeddings could have the power for more sophisticated programming tutors by identifying conceptual misconceptions that are reflected in student code structure (Haldeman, 2021).

Security applications are another potential direction. The combination of sequential and graph based attention could do better vulnerability detection by modeling both syntactical vulnerability patterns and propagation paths of patterns in call graphs (Wu & Zou, 2022). Similarly the embeddings may boost malware Using Hierarchical Structure Formal analysis Using hierarchical organization of malicious code components (Guo et al., 2025).

## 7.3 ETHICAL CONSIDERATIONS

The potential of misuse in automated vulnerability discovery and special safeguards required for exploiting generation (Vemuri et al., 2023). Model's Ability to Learn and reproduce code patterns also introduces copyright issues while training on Open Source repositories with different licenses (Gao et al., 2024).

For example, the model may inherit and propagate stylistic choices or patterns of architecture that reflect demographic imbalances in the software development world (Park et al., 2025). models (potentially reduced accountability in high stakes application) (Das & Rad, 2020).

# 8 CONCLUSION

The hierarchical cherry-picking of the code embedding system with multi-level attention Research into mechanisms provides major breakthrough in reinforcement learning state representation for code related task.

# 9 THE USE OF LLM

We use LLM polish writing based on our original paper.

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
