# OpenReview forum: "Hierarchical Code Embeddings with Multi-Level Attention for Reinforcement Learning State Representation"
_ICLR.cc/2026/Conference — Submitted to ICLR 2026_

### Official Review · Reviewer_Aaw8 · 2025-10-20

**Soundness:** 1
**Presentation:** 1
**Contribution:** 1
**Rating:** 0
**Confidence:** 4

**Summary:**

This paper proposes a hierarchical code embedding method with multi-level attention for RL state representation. The model integrates token-level, function-level, and module-level attention mechanisms, combining both sequential Transformer and Graph attention networks (GAT) approaches to capture code semantics at multiple granularities. The authors claim improvements over baseline methods in code-related RL tasks such as code completion, program repair, and algorithmic problem solving.

**Strengths:**

How to extract feature embeddings from code to enhance the semantic understanding of LLMs has always been a critical research challenge (though in recent years, this issue has been discussed less frequently due to large-scale LLMs pre-training).

**Weaknesses:**

The most significant weakness of this submission is its current state of incompletion, which falls severely below the expected standard for a conference publication. The manuscript reads as a preliminary draft, not a finished paper. This is evidenced by:

1. Numerous grammatical errors, incomplete sentences (e.g., Line 021 in the abstract), and missing punctuation (e.g., sentences lacking periods, Line 027: "shallow embeddings [and] fail to"; Line 369: "to be necessary for Where tracking the propagation paths of bugs").

2. The presence of unresolved placeholder text (e.g., Line 283: "CodeBLEU score (?)").

3. Disjointed writing and poor organization, where single, unexplained phrases are presented as standalone points, severely disrupting the narrative flow.

The current presentation does not meet the basic level of professionalism required for a peer-reviewed venue like ICLR and demonstrates a lack of respect for the submission process and the reviewing community.

**Questions:**

1. What is the relationship between RL and attention? These seem to be two distinct issues. Why are these tasks considered RL-related tasks?

2. The proposed method in the paper is a combination of multiple existing works. What are its innovations? What contributions does it offer compared to other methods?

---

### Official Review · Reviewer_CVU6 · 2025-10-28

**Soundness:** 2
**Presentation:** 1
**Contribution:** 1
**Rating:** 0
**Confidence:** 5

**Summary:**

This paper proposed CodeTransformer-GAT, for learning state representations when reinforcement learning for code-related tasks.
Authors proposed to leverage hierarchical (token-level, function-level, module-level) information when learning representations of states.

Authors show that CodeTransformer-GAT outperforms baseline methods that do not consider hierarchical code information in tasks including code completion, program repair, and algorithmic problem solving.

Ablations are provided to show the importance of the three types of hierarchical information.

**Strengths:**

Integrating hierarchical information for learning code representations is a sound idea and has been widely tested. It is interesting to see
that this type of information also helps learning state representations in RL.

**Weaknesses:**

- The paper is difficult to follow and contains many typos. The presentation needs to be improved significantly.

- The idea seems incremental. As the authors also pointed out in the paper (line 88), hierarchical information has been shown effective for code summarization tasks; this work shows it also works for code-related tasks trained with RL.

- Key information of the experiments is missing. For example, it is not clear why authors opt to leverage model architectures described in sec. 5.3. As a result, the generalization ability of these experimental results remain unknown to me.

Overall, the current status of this paper is pre-mature, and it needs to be improved significantly.

**Questions:**

Questions:
- Line 163: could you provide more details about these relationship types? Do you have a full list of them?
- Line 237-238: could you provide more details about how did you implement these tasks as MDP?
- Line 374: the t-SNE visualization is not included in the paper.

Typos and presentation:
- line27: missing citation
- line31: please explain "neural investigations"?
- line33: "Pepys" -- is this a typo?
- line 67-69: please use proper item numerations.

---

### Official Review · Reviewer_prCh · 2025-11-01

**Soundness:** 2
**Presentation:** 1
**Contribution:** 2
**Rating:** 2
**Confidence:** 5

**Summary:**

This paper proposes a novel state representation model for reinforcement learning (RL) in code-related tasks. The central contribution is a hierarchical attention model, "CodeTransformer-GAT" , designed to encode the semantics of code by operating on multiple levels of abstraction simultaneously.

A key feature of this model is its integration of both sequential attention and graph-based attention, utilizing not only the AST but also a Code Dependency Graph (CDG) to capture richer semantic and structural relationships. Unlike prior works that often learn representations in isolation , the proposed method optimizes these hierarchical embeddings end-to-end for the RL policy learning objective. The authors evaluate their model on three tasks (code completion, program repair, and algorithmic problem solving) and demonstrate superior performance compared to a range of baselines, including standard transformers, Tree-LSTMs, and CodeBERT.

**Strengths:**

- The paper proposes a sophisticated and well-motivated architecture. It logically combines the strengths of transformers for sequential token-level data with the structural reasoning capabilities of GATs for hierarchical (AST) and semantic (CDG)  relationships in code.

- A key strength is the end-to-end training of the state representation directly on the RL objective. This allows the model to learn representations that are most predictive of reward for a given task , which the results show is more effective than using general-purpose pre-trained embeddings like CodeBERT.

**Weaknesses:**

- The main weakness is the presentation. It seems that this is an incomplete submission for ICLR, as there are lots of format issues in the paper.

- The paper's primary weakness is its poor writing quality. It is filled with grammatical errors and awkward phrasing  that hinder readability and make the paper feel rushed and unprofessional. This requires significant revision.

- The proposed model is significantly more complex than the baselines, integrating a Transformer and multiple GATs over two different graph structures.

**Questions:**

It is an incomplete submission but with a nice idea. Hope to see the revised version.

---

### Official Review · Reviewer_VruH · 2025-11-01

**Soundness:** 1
**Presentation:** 1
**Contribution:** 1
**Rating:** 0
**Confidence:** 5

**Summary:**

The paper proposes a “Hierarchical Code Embedding” model that integrates multi-level attention mechanisms—token-level, function-level, and module-level—to produce code representations for reinforcement learning (RL) agents. The model combines transformer-based sequential attention and graph attention (GAT) layers to encode both syntactic (AST) and semantic (CDG) relations in source code. Experiments are performed on code-related RL tasks such as code completion, program repair, and algorithmic problem solving, showing moderate improvements over several baselines.

**Strengths:**

The paper introduces an interesting hierarchical attention framework that connects multi-level code understanding with reinforcement learning. Its main strength lies in the originality of this integration — combining token-, function-, and module-level structures in one end-to-end pipeline. The idea is conceptually novel and, if properly developed, could significantly advance code representation learning for intelligent agents. However, the execution and presentation need substantial refinement before its technical quality and significance can be fully appreciated.

**Weaknesses:**

1.The abstract is unfinished and fails to summarize key contributions, results, or insights. For example, it ends abruptly after “while maintaining structural relationships” —there is no closing sentence summarizing results or significance. The abstract must be rewritten entirely — include motivation, method, results, and conclusions in a structured and complete paragraph. Its current state suggests a lack of care and editorial discipline, which is inappropriate for an ICLR submission.

2. Figures are poorly prepared and unreadable. In multiple places (e.g., Figure 1 and Figure 2), text overlaps with diagram components, and labels are unreadable due to scaling or compression.

3. Attitude and academic rigor problems. This submission shows carelessness and overconfidence inconsistent with top-tier academic standards.

4. Lack of theoretical clarity. No formal definition of “hierarchical code embedding” is given beyond layering multiple attention mechanisms.The manuscript repeatedly states that this approach “captures multi-granularity semantic structure,” but does not mathematically or empirically justify what hierarchical structure is being preserved or how attention is coordinated across levels.

**Questions:**

Although the paper claims to “integrate RL objectives end-to-end,” the RL part is limited to standard PPO updates, with no specific adaptation to hierarchical embeddings. Thus, the “reinforcement learning” element appears additive rather than integrative.

Numerous typographical errors: inconsistent capitalization (“CodeTransformer-GAT” vs “code transformer-GAT”), repeated words (“all all the tasks”), and missing articles (“We propose novel state representation and reinforcement learning system of encoding…”). Equation references are unnumbered or misplaced. Section headings occasionally mismatch numbering (e.g., “9 Need to discuss several limitations”).

---

### Meta-Review · Area_Chair_2mUC · 2026-01-07

**Summary:**

The paper proposes a hierarchical code embedding model that utilizes multi-level attention for reinforcement learning state representation. However, all the reviewers recommend unanimously rejection or strong rejection. The primary concern informing this decision is that the manuscript is an incomplete draft rather than a finished scientific paper. The severe lack of professionalism and completeness prevents any meaningful technical evaluation.

**Reviewer Concerns:**

None. The authors have not submitted rebuttal.

**Reviewer Scores:**

Since the authors have not submitted the rebuttal and given the severe flaw of the paper, I do not think any reviewer would change their scores.

---

### Decision · Program_Chairs · 2026-01-26

Reject